# A Meta-Analytical Review of Gender-Based School Bullying in Spain

**DOI:** 10.3390/ijerph182312687

**Published:** 2021-12-01

**Authors:** Sandra Feijóo, Raquel Rodríguez-Fernández

**Affiliations:** 1Department of Social Psychology, Basic Psychology, and Methodology, Universidade de Santiago de Compostela (USC), 15782 Santiago de Compostela, Spain; sandra.sanmartin@usc.es; 2Department of Methodology of Behavioral Sciences, Faculty of Psychology, Universidad Nacional de Educación a Distancia (UNED), 28040 Madrid, Spain

**Keywords:** bullying, gender, LGBT-phobia, prevalence, meta-analysis

## Abstract

School bullying continues to be one of the main challenges for the education community. Current research indicates that Lesbian, Gay, Bisexual, Transgender, Transsexual, and other LGBT+ people suffer the highest rates of bullying, while other studies suggest that this bullying does not occur based on the victim’s actual sexual orientation or gender identity, but because they do not fit into the traditional gender roles. The aim of the present study was to carry out a meta-analytical study on the prevalence of gender-based bullying against LGBT+ schoolchildren and adolescents in Spain. Methods: The review was carried out following the recommendations of the PRISMA group and allowing us to identify a total of 24 studies. All of these studies were published since 2008, and most of them conducted cross-sectional survey-type research. It was also found that the instrument used to assess bullying varied greatly among studies, resulting in an enormous heterogeneity of research on this topic. Different meta-analyses were carried out according to the profile of involvement in bullying: victimisation, perpetration, and observation. In addition, three target populations were detected in the victimisation research: the general population, pre-identified bullying victims reporting the reasons behind the victimisation, and LGBT+ people. Results: The meta-analyses conducted with R have estimated the prevalence of observation of gender-based school bullying in Spain at 77.3%, perpetration at 13.3%, and victimisation at 8.6% among the general population. When the research focuses on previously identified victims, the rate was 3.6%, while if LGBT+ people are approached directly, the percentage increases to 51%. Conclusions: These rates reveal the need to develop specific preventive strategies in schools. Greater awareness of affective-sexual diversity and respect for those who do not conform to traditional gender roles should be promoted.

## 1. Introduction

Bullying has been defined as deliberate aggression or intentional harm-doing carried out by one or several people repeatedly and over time in an interpersonal setting characterised by an imbalance of power, either real or perceived [1]. Research has suggested that any person or group different in some way from “the majority” is particularly vulnerable to victimisation [2,3,4,5]. There is even research that refers to Lesbian, Gay, Bisexual, Transgender, Transsexual, and other LGBT+ people as those who suffer the highest rates of school bullying and harassment in general [6,7]. In this sense, a person can be repeatedly exposed to exclusion, isolation, threats, insults, and physical aggression by those who use homophobia, sexism, and other values associated with heterosexism as justification [8].

There is a conceptual debate about how to define bullying specifically among LGBT+ people [9]. One of the key aspects of why it is so difficult to name this issue is that this bullying does not occur due to the true sexual orientation or identity of the victims, but because they do not fit into the traditional gender roles in some way [5,10]. Moreover, some authors have suggested that it could be included in the framework of gender-based violence [9,11], although in the Spanish context, it is only legally recognised as such when a man mistreats a woman with whom he has or has had a sentimental relationship [12]. Examples of people who are not perceived or represented by normative gender patterns would be feminists, who have often been called “lesbians”, or men who engage in activities considered “unsuitable” for their sex who are labelled as “fags”, especially by other men and regardless of the orientation of their sexual desire [8,13]. In addition, parents of LGBT+ children may also experience homophobic bullying, just as there is a sort of “contagion” of the stigma towards friends of LBGT+ people or even those who defend victims of homophobia [8]. Although different bullying research has used names linked to homophobia such as lesbophobia, biphobia, transphobia, or LGBT-phobia [9,14], the present research will use the term “gender-based bullying” in order to reflect the various theories that highlight the importance of traditional gender roles in this context [5,8,9,10,11,13].

From a global perspective, the Rainbow Europe website currently ranks Spain in eighth place among 49 Eurasian countries in which LGBT+ rights have been achieved, with an estimated 65% in the scope of human rights for this group, explicitly mentioning Education as one of the areas showing the greatest progress, thanks to the Organic Law 8/2013 of 9 December for the improvement of educational quality [15]. However, with regard to the school setting, in particular, Generelo and Pichardo [14] published a report pointing out the lack of institutional and academic attention to LGBT+ violence, while other publications have generalised the problem to all institutions or even to the Spanish society as a whole, characterising it as sexist and homophobic [16,17]. In this sense, a report published in 2015 found that 88% of students in secondary education reported having witnessed, at least once, taunts and insults related to sexual orientation and/or gender identity, such as “faggot”, “dyke”, or “tomboy” [18]. As a counterpoint, another 2016 report noted that only 3.2% of victims of bullying and 4.2% of victims of cyber-bullying perceived that they had been bullied due to their sexual orientation [19].

To sum up, it can be stated that gender-based school bullying is a problem that is difficult to conceptualise, all of which is compounded by the presence of contradictory information regarding its prevalence in Spain. However, no study seems to have been carried out to synthesise the knowledge about this type of bullying in our country. Therefore, the main objective of the present study was to carry out a meta-analytical review of gender-based bullying at schools in the Spanish context.

## 2. Materials and Methods

The meta-analytical review was conducted on the basis of the recommendations provided by the PRISMA group [20], Botella and Gambara [21], and Botella and Sánchez-Meca [22] but was not registered in the PROSPERO database (International Prospective Register of Systematic Reviews).

Three dimensions were defined for the bibliographic search: the problem to be studied (bullying), the population under study (the LGBT+ community), and a spatial dimension (the Spanish context). The terms used in the search refer to each of these dimensions in both Spanish and English in an attempt to be as exhaustive as possible, given the aforementioned terminological difficulties. The terms of the same dimension have been used with the connector “OR” or equivalent depending on the database used, while the dimensions have been linked with “AND” or equivalent. Terms related to bullying were searched for in the keywords field, while those related to the population could be anywhere in the text in an attempt to cover research that had included LGBT+ people, although not as the main focus of the study. The terms for bullying were: Bully*, Cyberbull*, Harass*, Bulli*, Violen*, Aggress* and Perpetrat*; with the Spanish equivalents Acos*, Ciberacos*, Hostiga*, Ciberbull*, Bulli* and Agres*. For the LGBT+ community, the terms were: Homo-phob*, Gender, “sexual orientation”, LGB*, Trans*, and Queer; with the Spanish equivalents Homofob*, Género, “orientación sexual”, LGB*, Trans* and Queer.

The databases Scopus, Dialnet, and Eric ProQuest, were used for the review, as well as the meta-search engine EBSCO Host. This systematic review was conducted between August and September 2020. The use of numerous search terms, some of them of an ambiguous nature (such as Trans*), resulted in the return of a wide range of results in the databases. Of these results, 110 were identified as being of potential interest to the present study. The inclusion criterion was that the research addressed school bullying or harassment at non-university levels with reference to gender, sexual orientation, or identity, and the exclusion criterion was that the target population did not belong to the Spanish territory. No time criterion was used so that all studies found were included regardless of their year of publication. Furthermore, in order to try to detect as many publications as possible, a manual review of the journals, Journal of Homosexuality, Aggressive Behavior, and International Journal of Bullying Prevention, was conducted. The Researchgate profiles of José Ignacio Pichardo Galán and Lucas Platero were also consulted, and the snowball method was used to identify other data sources from those detected during the review. This allowed for the identification of 27 additional documents. All records were checked by the first author, and no automation mechanism was used. A flow chart to illustrate the process of search and selection of studies to be included in the meta-analysis is presented in Figure 1.

As for the coding of the information, the following characteristics of the studies were collected: the autonomous community(ies) in which the study was conducted; the size of the sample (*n*) whose rate is reported; the instrument used to identify bullying; the time frame and frequency needed to consider it to be bullying and not just isolated violent conduct; the rates of gender-based bullying; and finally other data of interest about the study that could facilitate the interpretation of the results. When the documents did not explicitly include the rates, but provided sufficient information to calculate them (i.e., the absolute frequency), this calculation was made. On the other hand, when several rates of behaviours that may constitute bullying were provided, but not an overall rate, a conservative choice was made to include only the highest rate (not the aggregate because it could cause an overlap of the sample involved). Table 1 presents a summary of the most relevant information from the studies analysed. In addition, the characteristics of some of the studies analysed seemed to indicate that they shared the same database, so in order not to duplicate the results, only the first published paper was included (therefore, as indicated in the flow chart, three studies were excluded). These studies were Martxueta [23], Orue et al. [24], and Larrain and Garaigordobil [25]. 

The studies found had not only addressed involvement in bullying as a victim, but some had also investigated the observation of bullying of LGBT+ people or even the perpetration of bullying towards them. Thus, databases differentiated according to the type of involvement reported (victimisation, perpetration, observation) were created using Microsoft Excel 2019. In terms of victimisation, three clearly differentiated target populations were detected: the general population, LGBT+ people concretely, and victims of bullying who reported the reasons behind their victimisation and these reasons were gender-based (specifically, LGBT-phobic). Given the apparent variability of these populations and the prevalence associated with them, it was decided to conduct several meta-analyses based on these different target populations and not only on the type of involvement in bullying. According to the recommendations of Valentine et al. [26], even if a small number of studies are included, it is still relevant to conduct a meta-analysis. Thus, a total of five meta-analyses were conducted with different studies: (1) one with studies reporting observation (*n* = 5); (2) one with studies addressing perpetration (*n* = 6); (3) one with gender-based victimisation reported by the general population (*n* = 8); (4) one with victimisation prevalence among LGBT+ people (*n* = 9); (5) and finally, one with prevalence reported by victims of bullying indicating gender-based reasons behind the victimisation (*n* = 5). The study of Sastre et al. [19] was included in both the meta-analysis with the general population as well as in the victims one, as it provided data on both kinds of samples.

The R programming language [27] was used to synthesise the results and produce descriptive graphs, specifically the “meta” [28] and “metafor” [29] packages. Random-effects models were used to account for the heterogeneity of the various studies included in each meta-analysis and also to have greater generalisability of the results [30]. Following Viechtbauer’s recommendations [29], direct proportions were used as effect sizes when the observed proportions identified in the studies were between 0.20 and 0.80, as was the case for studies conducted specifically with LGBT+ people. On the other hand, logit transformations were used when proportions smaller than 0.20 were manipulated, such as those found in studies with the general population or with victims of bullying.

**Table 1 ijerph-18-12687-t001:** Characteristics of the studies selected for inclusion in the meta-analysis.

STUDY	REGION	SAMPLE	SAMPLE AGE & SEX	INSTRUMENT	TIME FRAME	FREQUENCY	RATE
1. Gualdi et al., 2008 [31]	Madrid	152 students	Not reported	Self-reported item	Last academic year	At least once	73% victimisation
2. Garchitorena, 2009 [32]	Whole Spain	325 LGBT+ students	$\bar{x}$ = 20.9; 45.5% girls	Self-reported item	Whole lifetime	At least once	56.8% victimisation
3. INJUVE, 2011 [33]	Whole Spain	1411 participants	Between 15 and 29; 49% girls	Self-reported item	Whole lifetime	At least once	75% observation
4. Generelo, 2012 [34]	Whole Spain	653 LGBT+ students	Under 25; 34% girls	Self-reported item	Whole lifetime	At least once	71% victimisation
5. López et al., 2013 [16]	Whole Spain	762 LGBT+ participants	Different ages; 41% women	Self-reported item	Whole lifetime	At least once	76% victimisation
6. Pichardo et al., 2013 [13]	Madrid and Canary Islands	4636 students	Between 11 and 19 50.21% girls	Self-reported item	Whole lifetime	At least once	16% victimisation 30.5% perpetration 83.2% observation
7. FELGTB, 2013 [35]	Whole Spain	1000 LGBT+ students	Not reported	Self-reported item	Whole lifetime	At least once	65.3% victimisation
8. Martxueta & Etxeberria, 2014 [36]	Basque Country	119 LGBT+ students	Different ages; 26.89% women	*Olweus Bully/Victim Questionnaire*	Last two months	At least once	30.25% victimisation
9. Pichardo et al., 2015 [18]	Madrid, Canary Islands and Andalusia	3236 students	Age not reported 47.1% girls	Self-reported item	Whole lifetime	At least once	12% victimisation
10. Fundación Mutua Madrileña & Fundación ANAR, 2016 [37]	Whole Spain	550 victims	Not reported	Self-reported item	Whole lifetime	At least once	2.7% victimisation
11. Benítez-Deán, 2016 [38]	Community of Madrid	5605 students	Secondary students (age not reported), 49.07% girls	Self-reported items	Whole lifetime	At least once	3.04% victimisation 59.68% observation
12. Sastre et al., 2016 [19]	Whole Spain	21,487 students	Between 12 and 16 years 48.3% girls	EBIPQ and ECIPQ	Last two months	At least once per week	0.3% victimisation 0.32% perpetration
13. Fundación Mutua Madrileña & Fundación ANAR, 2017 [39]	Whole Spain	365 victims	Not reported	Self-reported item	Whole lifetime	At least once	2.9% victimisation
14. Generalitat Valenciana, 2017 [40]	Valencian Community	2484 victims	Not reported	Bullying reports intervened by school management teams	2015–2016 school year	At least once	5.39% victimisation
15. Gutiérrez-Barroso & Pérez-Jorge, 2017 [41]	Canary Islands	3723 students	Secondary students (age not reported), 50% girls	Self-reported item	Last year	At least once	14.7% victimisation 5.9% perpetration
16. Elipe et al., 2018 [42]	Andalusia	69 LGBT+ students	Not reported for LGBT+ subsample (Overall: $\bar{x}$ = 14.9, 49.4% girls)	EBIPQ	Last two months	At least once per week	45.4% victimisation
17. Fundación Mutua Madrileña & Fundación ANAR, 2018 [43]	Whole Spain	247 victims	Not reported	Self-reported item	Whole lifetime	At least once	3.2% victimisation
18. Orue & Calvete, 2018 [44]	Basque Country	791 students	$\bar{x}$ = 13.96 years 43.61% girls	*Escala de acoso escolar homofóbico*	Last month	At least once	79% observation 23.2% perpetration
19. Aparicio-García et al., 2018 [45]	Whole Spain	233 LGBT+ students	Not reported for LGBT+ subsample (Overall between 14 and 25 years)	Self-reported items	Whole lifetime	At least once	42.9% victimisation
20. Kualitate Lantaldea & ALDARTE, 2018 [46]	Basque Country	107 LGBT+ participants	Not reported	Self-reported items	Whole lifetime	At least once	45% victimisation
21. Albaladejo-Blázquez et al., 2019 [47]	Valencian Community	1723 students	$\bar{x}$ = 13.39 years 49% girls	*The Homophobic Verbal Content Bullying* of HCAT	Last week	Three or more times	25.31% victimisation 29.48% perpetration
22. Rodríguez-Hidalgo & Hurtado-Mellado, 2019 [48]	Andalusia	820 students	$\bar{x}$ = 14.87 years 51.7% girls	Homophobic EBIPQ	Last two months	At least once per week	23% victimisation
23. Martínez-Gómez et al., 2019 [49]	Valencian Community	87 students	$\bar{x}$ = 13,34 years 50.6% girls	*Escala de Vivencias de discriminación*	Whole lifetime	At least once	13% victimisation 10.4% perpetration 89.5% observation
24. Garaigordobil & Larrain, 2020 [50]	Basque Country	219 LGBT+ students	Not reported for LGBT+ subsample (Overall: between 13 and 17 years, 52.6% girls)	*Escala de Screening de acoso entre iguales*	Whole lifetime	At least several times	25.1% victimisation

## 3. Results

Of the studies included in the different meta-analyses, one was carried out with a sample from three different Autonomous Communities (Community of Madrid, Canary Islands and Andalusia); one with a sample from the Community of Madrid and the Canary Islands; two were carried out only in the Community of Madrid; one in the Canary Islands; four in the Basque Country; three in the Community of Valencia; and two in Andalusia. The rest (*n* = 10) were carried out in an attempt to cover samples distributed geographically throughout Spain. All the studies found have been published since 2008, and 15 of them specifically only in the last 5 years. In terms of sample size, 10 of the studies had a sample size of less than 400, the smallest *n* = 69, and the largest *n* = 21,487. The most common type of study was a cross-sectional analysis using a survey-type research tool. It was also found that the measurement instrument used was different in almost all cases. Only eight papers used multi-item tools, three studies used the *European Bullying Intervention Project Questionnaire [EBIPQ]*, another used the *Olweus Bully/Victim Questionnaire*, another used *The Homophobic Verbal Content Bullying* subscale of the Homophobic Content Agent Target scale [HCAT], another used the *Escala de acoso escolar homofóbico* (Homophobic Bullying Scale), another the *Escala de Vivencias de Discriminación debido a la Orientación o la Identidad Sexual* (Experiences of Discrimination due to Sexual Orientation or Sexual Identity Scale), and the *Escala de Screening de Acoso entre Iguales* (Peer Bullying Screening Scale) was used by the eighth study. There was one study that employed an entirely unique methodology, consisting of assessing the reports of situations of alleged bullying intervened by school management teams. All the other studies used one or more items elaborated by the authors’ ad hoc, in which each participant self-reported having experienced, committed, or observed the item described (bullying in general or a specific behaviour). The materials used were not the only thing that differed from one study to another, as the definitions of the timing of having experienced bullying and what was understood by bullying were also not homogeneous from one study to another. However, in order to retain as much information as possible, it was decided to include all identified studies in the appropriate meta-analysis. This variability between studies is reflected in the meta-analyses, all of which have an I^2^ index (heterogeneity) between 78% and 100%. 

First, a meta-analysis of the results on the observation of gender-based bullying was carried out. This involved an aggregate *n* of 12,682 participants, spread across six different studies. The total prevalence extracted by the random effects model was 77.3%, with a 95% confidence interval (CI) of between 65.9% and 85.7%. Figure 2 shows the dispersion of the data extracted from each of the studies, as well as the information associated with the meta-analysis.

A second meta-analysis was then conducted on the results of gender-based bullying perpetration, with an aggregated sample of 32,447 participants from six different studies. The total prevalence extracted following the random effects model was 13.35%, with a 95% confidence interval (CI) of between 4.8% and 31.8% (see Figure 3).

On the other hand, for the synthesis of the victimisation results, a meta-analysis was carried out with the results obtained from the general population. This involved an aggregate sample of 41,317 participants from 8 different studies. The total prevalence extracted following the random effects model was 8.6%, with a 95% confidence interval (CI) of between 4.5% and 15.9%. Figure 4 shows the dispersion of the data extracted from each of the studies, as well as the information associated with the meta-analysis.

The following meta-analysis was carried out with the results obtained from LGBT+ people, either because of their sexual orientation or gender identity. An aggregate sample of 3487 participants across nine different studies was used. The total prevalence extracted by the random effects model was 51.1%, with a 95% confidence interval (CI) of between 39.9% and 62.3% (see Figure 5).

Finally, the last meta-analysis conducted included the five studies that addressed the gendered motivations behind the bullying of pre-identified victims. This analysis had an aggregate sample of 5644 participants and revealed an overall prevalence of 3.6%, with a 95% confidence interval (CI) of between 2.6% and 5%. This is the analysis with the lowest, but still high, heterogeneity (I^2^ = 78%). Figure 6 details the information from this analysis.

Regarding the detection of publication bias, no formal analysis has been applied in the present study for this purpose, as Egger’s regression test is discouraged for meta-analyses of 25 or fewer studies [51].

## 4. Discussion

The present meta-analytical study had been proposed with the main objective of systematically reviewing and meta-analysing the research carried out on the prevalence of gender-based bullying among schoolchildren and adolescents in the Spanish context. In this sense, diverse sources of information with even more diverse realities have been found. Violence not only has consequences on those who suffer it and those who perpetrate it, but also on those who observe it [52], so it was of interest to know the reality of victims and perpetrators, but also that of observers. The results of the present study have established the prevalence of observation of gender-based bullying in Spain at 77.3%, perpetration at 13.35%, and victimisation at 8.6%. However, it should be taken into account that the latter figure covers the entire population, whereas when the research focuses on the previously identified victims of bullying, it is estimated at 3.6%. Furthermore, if studies focus on people with sexual orientations or gender identities other than the “norm”, bullying rises up to 51%. Some studies also reported online bullying or cyberbullying, but their samples were very disparate to be included in the same meta-analysis: Generelo [34] reported victimisation among LGBT+ participants, while Fundación Mutua Madrileña and Fundación ANAR [37,39,43] and Sastre et al. [19] reported gender-based bullying among previously identified victims. Regarding the places where bullying seems to occur most frequently, those are the spaces in which students spend more time in contact, such as the classroom during classes, the playground, and particularly the classroom and the corridors of the school during the time between classes [18].

Research on gender-based bullying in Spain is limited to little more than the last dozen years. The first study found addressing this issue was published in 2008 [31]. Until 2013, almost one study per year was published, with a substantial increase from 2016 onwards. There were both specific research as well as studies on bullying in general that addressed the motivations behind the bullying, the groups that suffer it or the potential homophobic content. This homophobic content is present regardless of whether or not the person who had suffered the bullying did actually identify as LGBT+ [48]. This highlights the difficulties in addressing gender-based bullying. On the other hand, it is worth noting the high number of studies (*n* = 29) that were discarded because despite appearing with the identifier “gender”, they actually explored “sex” differences between girls and boys. The way in which both constructs are understood by both researchers and study participants may be a major bias in the research. While sex is a biological characteristic associated with physical and physiological traits, gender is a social construct related to the roles, behaviours, and identities associated with a particular sex [53]. Sex and gender are often binarily categorised as “female” or “male” and are sometimes used interchangeably due to their complex relationship [54]. 

If the conceptualisation of gender is already controversial, it is even more controversial to address and define gender-based bullying. The studies included in the meta-analysis addressed homophobic or transphobic bullying, or even simply attacks on non-heterosexual people. Comparing LGBT+ people within the same sample with cis-heterosexual people (i.e., those whose gender identity matches their physiological sex) finds that LGBT+ people experience higher rates of bullying [42,50]; however, it cannot be guaranteed that the motivation behind the bullying is LGBT-phobic. On the other hand, focusing on LGBT+ victims also ignores that there are cis-heterosexual people who suffer homophobic attacks because they do not fit traditional gender roles [5,10]. Some of the reviewed studies have openly addressed homophobic bullying suffered by people regardless of their real orientation or identity [48]. Special mention should be made of the study of Donoso-Vázquez et al. [55] that was excluded during the Eligibility phase of the search as it was focused on the online setting and not schools. Although it was not included in the meta-analysis, it constituted a true analysis of gender-based violence, exploring a wide range of motivations linked to heterosexism, such as transgressing female sexual normativity or the heteronormative beauty canon.

In addition to the difficulties present in the study of bullying in a specific group, research on bullying, in general, has an inherent series of obstacles when carrying out a comparative and integrative analysis of prevalence due to the diversity of samples, instruments and methodologies used in different investigations [56,57,58]. In this sense, in the present review, we found immense diversity—from a study focused solely on the verbal content of bullying during the last week [47], and another on LGBT+ people’s lifetime bullying assessed retrospectively [36], to a review of the motivations behind bullying cases intervened in schools [40], or a study using a multi-item tool adapted to ask about homophobic bullying during the previous two months [48]. The type of behaviours that are recorded as bullying may have an impact on the reported rates, as when the same research studied different types of bullying suffered, the most frequent attacks were verbal, while physical aggressions were less frequent [32,33,38]. These results are similar to those frequently reported in the general literature on bullying [59].

On the other hand, Zych et al. [60] pointed out that the rates of cyberbullying found in local studies vary so substantially that it was advisable not to extrapolate them to the whole country or to other regions. Thus, the lack of specific bullying studies in certain areas of Spain may pose a major problem for the effective prevention of this issue. Specifically, the only Autonomous Communities that had carried out studies addressing the problem of gender-based bullying were the Community of Madrid [13,18,31,38], the Canary Islands [13,18,41], the Basque Country [36,44,46,50], the Valencian Community [40,47,49], and Andalusia [42,48]. Although several of the studies had attempted to access samples from the entire country [16,19,32,33,34,35,45], not providing data segmented by CCAA or not guaranteeing the representativeness of the different communities in the total sample, means that the lack of information in certain regions of Spain constitutes a score to settle by the research community.

Regarding the possible limitations of the present study, first of all, the controversial terminology used to describe the object of study should be mentioned, which makes it difficult to correctly identify the research carried out on the subject. At the same time, numerous documents have been detected through their citation in the studies identified in the databases (snowball effect) that did not use any words referring to gender-based bullying as key terms. On the positive side, this may indicate the transversality that this issue is adopting, but it may also imply that this meta-analytical review may have failed to identify relevant documents. In addition, many of the studies that were found came from the autonomous communities themselves or from associations, highlighting the need for greater involvement of the scientific community in researching this topic. Besides, future systematic reviews or meta-analyses could benefit from the inclusion of more databases (Web of Science, PsychINFO, etc.). Finally, although the great disparity at the conceptual level of the studies to be synthesised is a problem common to all research on bullying [56,57], the use of different time criteria, instruments, and methodologies means that the studies included in the same meta-analysis present great heterogeneity. In an attempt to minimise this limitation, the reported data have been calculated using a random effects model. This high heterogeneity reveals the need to detect which variables may be moderating the disparity in reported rates from one study to another. Future work should explore these moderating factors using meta-regression techniques if the number of available studies allows it. On the other hand, as it has already been highlighted by another meta-analytic review outside the Spanish context [58], it would be very useful for studies on bullying to provide sufficient information to estimate prevalence, since not all the studies identified at the outset were likely to be included in a meta-analysis. Furthermore, the research community is encouraged to further explore the motivations and causes behind bullying, rather than just reporting overall rates. Finally, there is also the need to conduct further research to analyse how the situation may have changed due to the COVID-19 pandemic, as the study of Vaillancourt et al. in Canada seemed to indicate that, while the pandemic may have mitigated bullying rates in general, the victimisation of gender-diverse and LGBT+  students remained higher than their gender binary or heterosexual peers [7]. 

## 5. Conclusions

The meta-analyses conducted with R have estimated the prevalence of observation of gender-based school bullying in Spain at 77.3%, perpetration at 13.3%, and victimisation at 8.6% among the general population. When the research focuses on previously identified victims that report gender-based reasons behind the bullying, the rate was 3.6%, while if LGBT+ people are asked directly about their bullying experiences, the percentage increases to 51%. The results of this meta-analytic review should have important implications for the study and intervention of gender-based bullying in Spain. The violation of human rights it represents and the impact it poses to both the social and individual well-being for one out of every two LGBT+ people suffering it should motivate the development of specific preventive strategies. It may be argued that greater awareness of affective-sexual diversity and respect for those who do not conform to traditional gender roles should be promoted in education, and society as a whole. 

## Figures and Tables

**Figure 1 ijerph-18-12687-f001:**
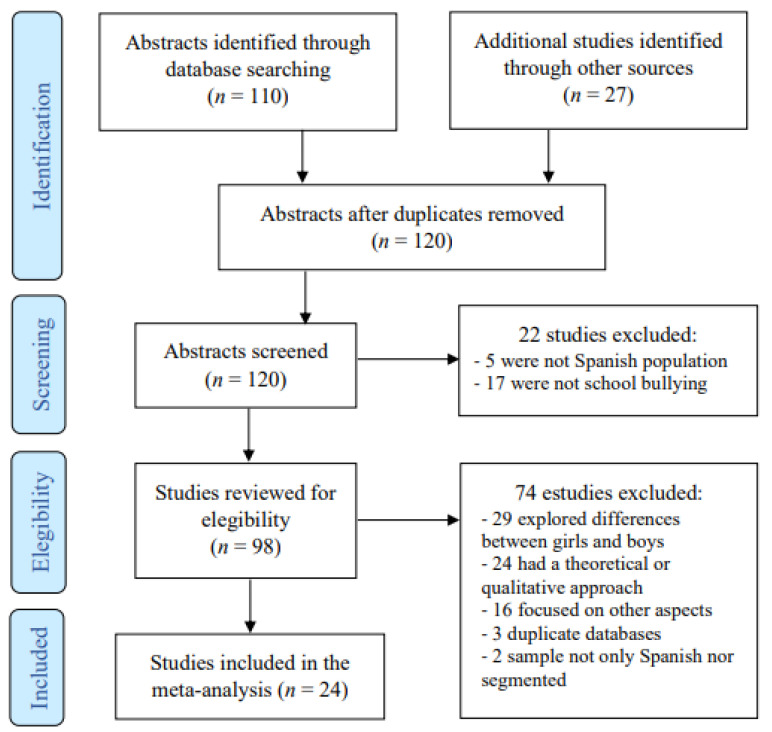
Flowchart of systematic search. Based on PRISMA 2009 Flow Diagram [20].

**Figure 2 ijerph-18-12687-f002:**
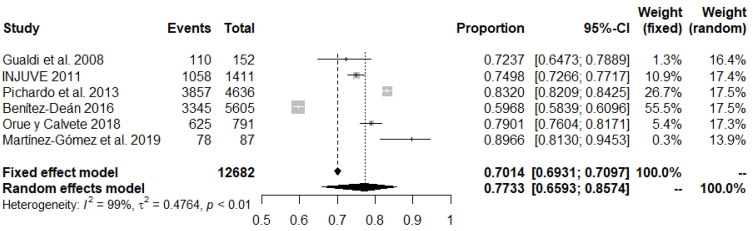
Forest plot of bullying observation studies.

**Figure 3 ijerph-18-12687-f003:**
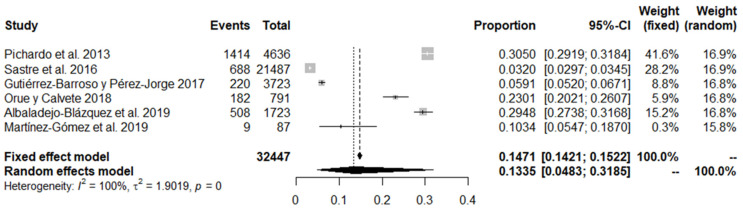
Forest plot of bullying perpetration studies.

**Figure 4 ijerph-18-12687-f004:**
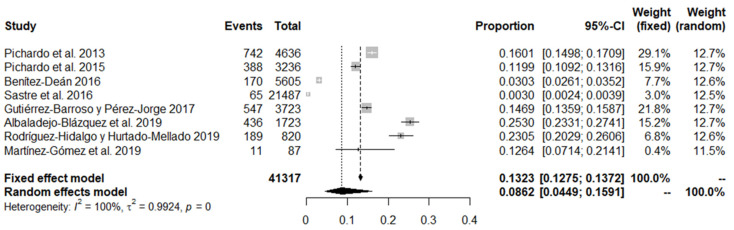
Forest plot of bullying victimisation among general population studies.

**Figure 5 ijerph-18-12687-f005:**
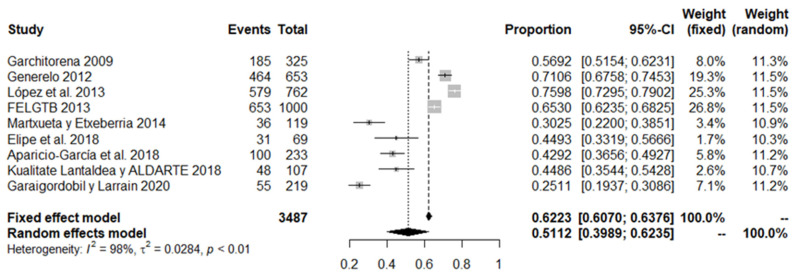
Forest plot of bullying victimisation among LGBT+ people studies.

**Figure 6 ijerph-18-12687-f006:**
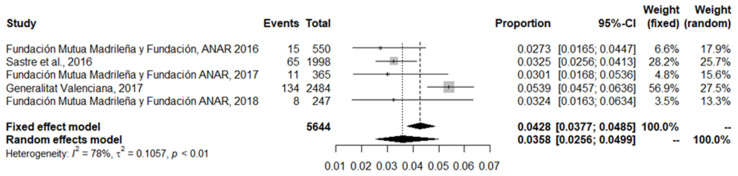
Forest plot of gender-based bullying victimisation among pre-identified victims’ studies.

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
