# Peer review of "A Meta-Analytical Review of Gender-Based School Bullying in Spain"

_ijerph, 2021, doi:10.3390/ijerph182312687_

Round 1
Reviewer 1 Report
The authors present an interesting systematic review on contextualized gender bullying in Spain. But the article in its current version has important scientific deficiencies that prevent its publication.
Below I show my suggestions for improvement so that the authors can develop them.
Title
The title must be modified, since the authors do not undertake a meta-analysis of all age groups, only of schoolchildren and adolescents. Therefore, the title must add the population under study and not be generic, as it leads to confusion.
Abstract
The abstract should begin with a statement about the rationale, before describing the objective of the study. The abstract should refer to the prism protocol
Introduction
Although the rationale is well established, the analysis of a greater number of variables that can influence bullying is lacking. These variables are indicated in the discussion section.
It is recommended to also include information on the school-adolescent context, since it is the age group in which the article focuses. It is important to talk about the places where bullying happens most: school playground, school hallways, entering and leaving the school center, etc.
Methodology
The authors used three databases for the search but have not used the largest Web of Science database, which brings together more than 170 million documents, compared to 70 million for Scopus or 7 million for Dialnet, so the review it could present a significant bias. Authors are suggested to search the Web of Science since they could find a greater number of scientific articles related to the subject of the review, and thus improve the quality of the meta-analysis.
I would like to ask the authors for the check of the PRISMA protocol indicating the fulfillment of the items and the place of location in the text.
Table 1 omits important information from the sample, such as the mean age of the subjects or sex and / or gender. Authors should incorporate this information to improve understanding of the results of each study.
Results
Authors are invited to create tables with the information that appears between lines 166 and 195. It is information that could be better visualized in tables.
Discussion
In line 238 again it should be indicated that the target population of the meta-analysis are young minors.
Although the authors make a good discussion, a discussion related to variables that can influence bullying, such as home schooling by parents, their educational level, physical activity practice, self-esteem levels, among others, is omitted. All these variables would be important to comment on both in the rationale and in the discussion.
Conclusions
This section should be improved since the authors are not concluding based on the objective of the research and based on the results obtained, in a certain way they are speculating. They should synthesize their main findings and list them in this section. Later they can offer their point of view but without concluding facts that they have not investigated. For example, they have not investigated anti-bullying plans in the centers and their effectiveness, so they cannot indicate that implementing these plans will mean a lower prevalence of bullying in the classrooms.
Authors are invited to withdraw the reference (Vaillancourt et al. (2021), bibliographic references should not be included in the conclusions section, since the authors should conclude on their findings and not on other studies. This reference can be moved to the rationale or discussion section, but not conclusions.
Study limitations and improvement proposals
Authors should incorporate the weaknesses of the study and offer potential readers new avenues for future research.
In general, the article is interesting and it is an important topic on a social level, however, it has many shortcomings in the scientific protocol that prevent it from being published. The authors are invited to make the indicated improvements and expand the review for more studies.
.
Author Response
Dear Reviewer,
Thank you for giving us the opportunity to submit a revised draft of our manuscript and for devoting time to provide feedback about our work. We have incorporated several changes based on your comments. Please find our responses below and the new additions marked with the Change Track in the manuscript. We have also adapted the references to the style of IJERPH, as they were previously in APA format.
We look forward to hearing from you in due time regarding our submission and responding to any further questions and comments you may have.
Sincerely,
The authors
Reviewer 1 comment 1: The title must be modified, since the authors do not undertake a meta-analysis of all age groups, only of schoolchildren and adolescents. Therefore, the title must add the population under study and not be generic, as it leads to confusion.
Answer: We have modified the title, so it is clear the review was focused on schoolchildren and adolescents. We have included this information in the abstract as well, when describing the aim of the present study.
Reviewer 1 comment 2: The abstract should begin with a statement about the rationale, before describing the objective of the study. The abstract should refer to the prism protocol
Answer: Following reviewer recommendations, abstract now begins with the rationale behind the study, and reference to PRISMA has been included.
Reviewer 1 comment 3: Although the rationale is well established, the analysis of a greater number of variables that can influence bullying is lacking. These variables are indicated in the discussion section.
Answer: We want to thank the reviewer for their comment about the rationale of the study. The lack of information on other variables is due to the articles reviewed not addressing any of the variables suggested by the reviewer, although we wanted to mention them in the Discussion section as we believe they are very relevant for a better understanding of bullying.
Reviewer 1 comment 4: It is recommended to also include information on the school-adolescent context, since it is the age group in which the article focuses. It is important to talk about the places where bullying happens most: school playground, school hallways, entering and leaving the school centre, etc.
Answer: In line with what happened with the variables referred to in the previous comment, this information was absent in the articles reviewed, with the exception of a single document, so we have included these results in the Discussion as part of the findings of the review.
Reviewer 1 comment 5: The authors used three databases for the search but have not used the largest Web of Science database, which brings together more than 170 million documents, compared to 70 million for Scopus or 7 million for Dialnet, so the review it could present a significant bias. Authors are suggested to search the Web of Science since they could find a greater number of scientific articles related to the subject of the review, and thus improve the quality of the meta-analysis.
Answer: The reviewer raises an important point and their suggestion is very pertinent and could certainly enrich our work, but we consider that with the databases used we found most of the existing studies in the Spanish population, so we can have a fairly accurate picture of the reality of this phenomenon in our country. We opted to use Dialnet as it is the database where most of the Spanish research is located, which was the study population for the present review. We also conducted a review in the database Eric, as it collects all the relevant journals for education, as the topic of study was school bullying. To complement the use of these databases, we have also used Scopus and the meta-search engine EBSCO Host, which hosts 375 full-text databases, a collection of 600,000-plus ebooks, trying to reach more documents that merely with Dialnet and Eric, but at the same time ensuring that these documents would be made open access from the authors' university network. However, we will certainly follow the recommendation to use Web of Science in future reviews we initiate.
Reviewer 1 comment 6: I would like to ask the authors for the check of the PRISMA protocol indicating the fulfillment of the items and the place of location in the text.
Answer: We have added the PRISMA checklist as attachment. Please note that the Location where item is reported corresponds to the document with activated change control.
Reviewer 1 comment 7: Table 1 omits important information from the sample, such as the mean age of the subjects or sex and / or gender. Authors should incorporate this information to improve understanding of the results of each study.
Answer: This information has been included in Table 1.
Reviewer 1 comment 8: Authors are invited to create tables with the information that appears between lines 166 and 195. It is information that could be better visualized in tables.
Answer: This information has been included in Table 1.
Reviewer 1 comment 9: In line 238 again it should be indicated that the target population of the meta-analysis are young minors.
Answer: Following reviewer recommendation, it has been specified that the meta-analysis was about bullying among schoolchildren and adolescents.
Reviewer 1 comment 10: Although the authors make a good discussion, a discussion related to variables that can influence bullying, such as home schooling by parents, their educational level, physical activity practice, self-esteem levels, among others, is omitted. All these variables would be important to comment on both in the rationale and in the discussion.
Answer: Variables such as home school by parents, level of education, physical activity, levels of self-esteem, etc. that may influence bullying were not included in either the rationale or the discussion as, unfortunately, they are all absent from the literature reviewed in our study. We only found one paper that did addressed places where discrimination took place, the results of which were included in the Discussion. On the other hand, several of the papers reviewed did indicate the type of behaviour suffered, so this aspect was also included in the Discussion.
Reviewer 1 comment 11: This section should be improved since the authors are not concluding based on the objective of the research and based on the results obtained, in a certain way they are speculating. They should synthesize their main findings and list them in this section. Later they can offer their point of view but without concluding facts that they have not investigated. For example, they have not investigated anti-bullying plans in the centres and their effectiveness, so they cannot indicate that implementing these plans will mean a lower prevalence of bullying in the classrooms.
Answer: The main findings of the research have been included in the Conclusion. While we wanted to retain our impressions of the implications for prevention, we have modified the wording to make it clear that these are the authors' inferences.
Reviewer 1 comment 12: Authors are invited to withdraw the reference (Vaillancourt et al. (2021), bibliographic references should not be included in the conclusions section, since the authors should conclude on their findings and not on other studies. This reference can be moved to the rationale or discussion section, but not conclusions.
Answer: The sentence including the mentioned reference has been moved to the end of the Discussion, and the wording has been adjusted to accommodate this sentence in its new location.
Reviewer 1 comment 13: Authors should incorporate the weaknesses of the study and offer potential readers new avenues for future research.
Answer: The limitations and future research proposals are addressed in the end of the Discussion, where Vaillancourt et al. (2021) has also been included as a recommendation for further research studying how Covid-19

Reviewer 2 Report
Please proof the manuscript carefully so you correctly and respectfully refer to those being bullied (the abstract refers to LGTB-phobic bullying when LGBT+ is appropriate). Writing style has numerous grammatical errors (for instance, "The review allowed to identify" [missing word], "being the most common cross-sectional survey-type research"). In lines 19-20, it is not clear to which specific rates you are referring when you discuss previously identified victims or LGBT+ people who are approached directly. Spain's Rainbow Europe ranking needs context (8th of how many European countries included?). What is the LO 8/2013 of 9 December mentioned in lines 62-63? Correct spelling in the chart on p. 3 is Eligibility (not Elegibility).
Author Response
Dear Reviewer,
Thank you for allowing us the opportunity to submit a revised draft of our manuscript and devote time to provide feedback about our work. We have incorporated several changes based on your comments. Please find our responses below and the new additions marked with the Change Track in the manuscript.
We look forward to hearing from you in due time regarding our submission and responding to any further questions and comments you may have.
Sincerely,
The authors
Reviewer 2 Comment 1: Please proof the manuscript carefully so you correctly and respectfully refer to those being bullied (the abstract refers to LGTB-phobic bullying when LGBT+ is appropriate).
Answer: We want to thank the reviewer for noticing this overlook on our part. Abstract now reads as follows: “bullying against LGBT+ schoolchildren and adolescents in Spain”. We have also checked the manuscript for more of these mistakes.
Reviewer 2 Comment 2: Writing style has numerous grammatical errors (for instance, "The review allowed to identify" [missing word], "being the most common cross-sectional survey-type research").
Answer: Both grammatical errors have been corrected and the text has been proofread again to identify further mistakes.
Reviewer 2 Comment 3: In lines 19-20, it is not clear to which specific rates you are referring when you discuss previously identified victims or LGBT+ people who are approached directly.
Answer: the sentences have been slightly modified to clarify to which population each rate belongs.
Reviewer 2 Comment 4: Spain's Rainbow Europe ranking needs context (8th of how many European countries included?).
Answer: 49 countries are included in Rainbow Europe’s ranking, not all of them are in the European Union, but are Eurasian (for example, United Kingdom, Norway, Moldova, and Turkey are also included). This information has been included in the text which now reads: “the Rainbow Europe website currently ranks Spain in eighth place among 49 Eurasian countries in which LGBT+ rights have been achieved”.
Reviewer 2 Comment 5: What is the LO 8/2013 of 9 December mentioned in lines 62-63?
Answer: “LO” is the acronym for “Ley Orgánica” (Organic Law). Full English name has been included in the text to facilitate understanding from an international audience.
Reviewer 2 Comment 6: Correct spelling in the chart on p. 3 is Eligibility (not Elegibility).
Answer: This typo has been corrected.
Round 2
Reviewer 1 Report
I greatly appreciate the changes made by the authors. However, I continue to insist that justifying that the DIALNET database brings together all the Spanish scientific information is an argument of little value. Most of the Spanish scientific production is published in international journals that are not indexed in Dialnet, hence all the information contained in this article presents a significant bias.
Dialnet is a Spanish database that uses Spanish as a vehicular language, but this does not imply that it contains Spanish scientific production.
For all these reasons, I consider that, despite the fact that the article has improved a lot since its initial version, I believe that its publication should be carried out in a lower impact journal, since its results can create confusion as they have a significant bias.
I invite the authors for future reviews to use the Web of Sciences and Scopus as two main mandatory databases to perform a systematic review or meta-analysis. Many of the variables that are not discussed are found in articles indexed in these databases. Dialnet, that's fine, but not for an article published in this scientific journal of the MDPI group, which requires a higher level of bibliographic search.
In the figures that I have provided to the authors in relation to the number of scientific documents contained in each of the databases, it is possible to observe the important bias that the article may have, out of a total of 170 million documents compared to 7 million that Dialnet owns. I encourage the authors to redo their work and incorporate the search in WoS, I am sure that the entire article will change and favor its acceptance and scientific dissemination of it.
Author Response
We want to thank the reviewer for reading our revised manuscript and providing further feedback. We would ask the reviewer to note that we are already using Scopus, in addition to the Eric ProQuest database and the EBSCO Host meta-search engine, and not just Dialnet. We will certainly follow your recommendation of including WoS as a mandatory database in the hope of improving future reviews, but including WoS on the present paper is not feasible at this stage. The systematic review was conducted between August and September 2020, so including a new database at this stage could end up being another source of bias, as much as we would try to control the publication period. Furthermore, although we consider that including the WoS database can improve the work, it should be borne in mind that several very current meta-analytical articles (all from this year) on subjects related to ours, and in journals with good impact index, do not include this particular database in their search (for example, Hu, Y., Bai, Y., Pan, Y., & Li, S. (2021). Cyberbullying victimization and depression among adolescents: A meta-analysis. Psychiatry Research, 305, 114198. doi:10.1016/j.psychres.2021.11419; Crane, C. A., Wiernik, B. M., Berbary, C. M., Crawford, M., Schlauch, R. C., & Easton, C. J. (2021). A meta-analytic review of the relationship between cyber aggression and substance use. Drug and Alcohol Dependence, 221, 108510. doi:10.1016/j.drugalcdep.2021.108; Tran, H. G. N., Thai, T. T., Dang, N. T. T., Vo, D. K., & Duong, M. H. T. (2021). Cyber-victimization and its effect on depression in adolescents: a systematic review and meta-analysis. Trauma, Violence, & Abuse, 15248380211050597)
We understand the concern about the possible bias caused by not including as many databases as possible, and particularly WoS, so we have explicitly raised this point in lines 355-356 (Change control activated).
We believe our present work still has strengths that would make it of interest to potential readers, mainly being a pioneering review on this topic in the Spanish context, but we understand the concerns raised about its possible limitations. Nevertheless, we want to note that we are grateful for the commitment of the reviewer to improve the quality of our work, and that the authors have lived this review as a learning opportunity to keep improving.
Reviewer 2 Report
Thank you for addressing the concerns noted.
Author Response
We want to thank the reviewer once again for devoting time to reading and improving our work.